# Fatigue Reliability Analysis Method of Reactor Structure Considering Cumulative Effect of Irradiation

**DOI:** 10.3390/ma14040801

**Published:** 2021-02-08

**Authors:** Bo Sun, Junlin Pan, Zili Wang, Yi Ren, Dariusz Mazurkiewicz, Małgorzata Jasiulewicz-Kaczmarek, Katarzyna Antosz

**Affiliations:** 1School of Reliability and Systems Engineering, Beihang University, No.37 XueYuan RD. Haidian, Beijing 100191, China; panjl@buaa.edu.cn (J.P.); wzl@buaa.edu.cn (Z.W.); renyi@buaa.edu.cn (Y.R.); 2Mechanical Engineering Faculty, Lublin University of Technology, ul. Nadbystrzycka 36, 20-618 Lublin, Poland; d.mazurkiewicz@pollub.pl; 3Faculty of Management Engineering, Poznan University of Technology, ul. Rychlewskiego 2, 60-965 Poznan, Poland; malgorzata.jasiulewicz-kaczmarek@put.poznan.pl; 4Faculty of Mechanical Engineering and Aeronautics, Rzeszow University of Technology, al. Powstańców Warszawy 8, 35-959 Rzeszow, Poland

**Keywords:** fatigue, reliability, neutron dose, irradiation effect, reactor structure

## Abstract

The influence of irradiation should be considered in fatigue reliability analyses of reactor structures under irradiation conditions. In this study, the effects of irradiation hardening and irradiation embrittlement on fatigue performance parameters were quantified and a fatigue life prediction model was developed. Based on this model, which takes into account the cumulative effect of a neutron dose, the total fatigue damage was calculated according to Miner’s linear cumulative damage law, and the reliability analysis was carried out using the Monte Carlo simulation method. The case results show that the fatigue life acquired by taking into account the cumulative effect of irradiation was reduced by 24.3% compared with that acquired without considering the irradiation effect. Irradiation led to the increase of the fatigue life at low strains and its decrease at high strains, which is in accordance with the findings of an irradiation fatigue test. The rate of increase in the fatigue life decreased gradually with the increase of the neutron dose. The irradiation performance parameters had a small influence on fatigue reliability, while the fatigue strength coefficient and the elastic modulus had a great influence on the fatigue reliability. Compared with the current method, which uses a high safety factor to determine design parameters, a fatigue reliability analysis method taking into account the cumulative effect of irradiation could be more accurate in the reliability analysis and life prediction of reactor structures.

## 1. Introduction

As efficient clean energy, nuclear power plays an irreplaceable role in confronting energy crises in the world today [1]. Some countries that have mastered nuclear power generation technology (such as France, the United States, China, etc.) view the active promotion of nuclear energy construction as an important energy construction policy [2]. In total, 31 countries operate 417 nuclear reactors and the total operating capacity has reached 370 GW, which is a new historic maximum [3]. At the same time, the reliability and safety of reactors are also receiving more and more attention because, once problems in reactors occur, they often cause disastrous consequences, such as the Fukushima accident, which resulted in all people living within a 20 km zone of the accident center being evacuated [4].

For the reactor structure, a design parameter such as material strength is determined by a high safety factor, which is conservative from the perspective of economy. The reactor structure is difficult to destroy directly (such as through a brittle fracture). However, a reactor has a long service life (more than 60 years) [5] and a long-term safe operation requirement. External loads, such as mechanical loads and environmental factors, lead to material degradation which may result in cumulative degradation failure over time. Fatigue is a common failure mode in reactor structures; for instance, the thermal fatigue of a T-junction piping system due to the continuous mixing of hot and cold water [6] or the fatigue caused by the mechanical load of a hold-down spring [7]. The fatigue load in a reactor includes periodic mechanical and thermal load, temperature difference and fluctuations of cooling water, as well as some fluctuations caused by startup, shutdown and emergency shutdown [8]. For reactor structures made of 304 or 316 stainless steels, the failure mechanisms of thermal fatigue and mechanical fatigue are obviously different, but there are no essential differences in the fatigue life prediction model [9,10,11]. The Coffin–Manson model can describe the relationship between strain amplitude and fatigue life and has been widely used in engineering [12]. For the fatigue problem of a reactor structure, Metzner [13] carried out a thermal fatigue assessment on a piping system in the reactor, established a thermal fatigue test database and proposed a thermal fatigue damage assessment process. Sudret [14] established a fatigue probability assessment framework for reactor components and provided an application case. Paffumi [15] used fatigue test data and a finite element analysis to evaluate the thermal fatigue damage of reactor 316 L piping.

However, the effect of irradiation is not generally considered in fatigue analyses of reactor structures. Some studies have carried out fatigue tests under different irradiation test conditions but obtained different (even contradictory) test conclusions [16,17,18,19]. Under certain irradiation test conditions, the fatigue life of the sample without irradiation is 1.5–2.5 times that of the sample with irradiation [16,17]. However, Zhong [18] found that the fatigue life of an irradiated sample was three times greater than that of a non-irradiated sample. Another study concluded that irradiation was beneficial to fatigue life at low strain amplitudes, but not at high strain amplitudes [19]. These different test results show that the effects of irradiation on fatigue are various and that there are many kinds of irradiation effects on fatigue. Irradiation can affect the tensile properties of materials. For austenitic stainless steels commonly used in the reactor structure, irradiation hardening and embrittlement are the main effects of irradiation, with irradiation hardening leading to increases of yield strength, ultimate tensile strength and other strength properties, and irradiation embrittlement leading to a decrease of elongation rate and a reduction of area and other ductility properties [20]. Fuller [21] quantitatively characterized the effect of irradiation on ultimate tensile strength, combining it with the Bäumel–Seeger uniform material law and Coffin–Manson model to predict fatigue life. In this study, only an irradiation hardening effect is considered. It is concluded that irradiation can reduce the fatigue life of the reactor structure. However, to improve the accuracy of fatigue life prediction under irradiation, it is necessary to quantitatively consider the effects of irradiation hardening and irradiation embrittlement at the same time.

Reliability is defined as the ability of a product to perform a specified function within a specified time and under a specified condition. In early reliability engineering, a probability measure was generally used to quantitatively characterize this capability. In practice, this probability is usually obtained through a statistical analysis of product failure time. The traditional method is based on statistics, which is suitable for mass-produced products but not for reactor structures. For reactor structures with high-reliability and long-life requirements, it is necessary to predict life and reliability. The design method with a higher safety factor can meet the requirements of a reactor design but it cannot predict the reliability quantitatively. More attention should be paid to the maintenance of long-term process functioning and performance, which is suitable for reliability analyses based on the physics of failure [22]. During the long-term operation of a reactor, there are degradation mechanisms such as fatigue, stress corrosion cracking and irradiation embrittlement. Irradiation-assisted fatigue is one of the keys factors affecting the operation reliability of reactor structures [8].

In this study, the Coffin–Manson model was used to characterize the relationship between fatigue life and strain. A basic fatigue life model was established for a reactor structure under irradiation. Based on the Coffin–Manson model, a fatigue life model taking into account the effects of irradiation on fatigue life was established by considering irradiation hardening and embrittlement. Further considering the cumulative effect of a neutron dose, the total fatigue damage was calculated according to Miner’s linear cumulative damage law. Utilizing the reliability definition, the corresponding reliability at different times was calculated by using the Monte Carlo simulation method. The described fatigue reliability analysis was able to quantitatively predict the reliability of the reactor structure during a long-term operation. Furthermore, this method provides a theoretical basis for the design and optimization of the reactor structure.

This paper proceeds as follows. Section 2 establishes a fatigue reliability model taking into account the cumulative effect of irradiation. Section 3 takes the core barrel flange as a case study on which to carry out the proposed fatigue reliability analysis considering irradiation effects. The influences of strain magnitude, neutron dose and material performance on the fatigue reliability are discussed in Section 4. Finally, Section 5 concludes this paper.

## 2. Materials and Methods

### 2.1. Fatigue Model Considering Irradiation Effect

#### 2.1.1. Fatigue Life Prediction Model

The fatigue life prediction is the main objective of a fatigue reliability analysis of materials or structures, which can quantify the “specified time” in the definition of reliability more clearly. A fatigue life prediction method based on S-N curves is often used in engineering, such as in Basquin’s equation [23]:(1)$$\frac{\Delta \varepsilon_{e}}{2}=\frac{\sigma_{f}^{\prime }}{E}{\left(2N_{f}\right)}^{b}$$

Currently, the Coffin–Manson equation is the most widely used fatigue life prediction method. It describes the relationship between fatigue life and plastic strain [24] and can be expressed as follows:(2)$$\frac{\Delta \varepsilon_{p}}{2}=\varepsilon_{f}^{\prime }{\left(2N_{f}\right)}^{c}$$

Combining the Coffin–Manson equation with Basquin’s equation, Manson and Hirschberg proposed a relation expression for fatigue life represented by the total strain [12], applicable to both low-cycle fatigue and high-cycle fatigue:(3)$$\frac{\Delta \varepsilon_{t}}{2}=\frac{\sigma_{f}^{\prime }}{E}{\left(2N_{f}\right)}^{b}+\varepsilon_{f}^{\prime }{\left(2N_{f}\right)}^{c}$$
where $\frac{\Delta \varepsilon_{t}}{2}$ is a total strain amplitude, $\sigma_{f}^{\prime }$ is a fatigue strength coefficient, *E* is the elasticity modulus, *b* is a fatigue strength exponent, $\varepsilon_{f}^{\prime }$ is a fatigue ductility coefficient, *c* is a fatigue ductility exponent and $N_{f}$ is fatigue life.

#### 2.1.2. Effect of Irradiation on Fatigue Parameters

After irradiation, the change of the performance parameter *P* with the neutron dose can be fitted by the following equation [20]:(4)$$P=A_{0}+A_{1}\left[1-exp(-d/d_{0}\right)]$$
where *d* is the neutron dose, $d_{0}$ is the irradiation model parameter, $A_{0}$ is the initial value for the performance parameter, $A_{1}$ is the irradiation coefficient for the performance parameter and *P* is the performance parameter under irradiation.

The effects of irradiation on fatigue life mainly manifest in two effects: irradiation hardening and irradiation embrittlement. Irradiation hardening is attributed to dislocation loops, stacking fault tetrahedrons, voids and other defects in the metal crystal, and leads to the improvement of the yield strength, tensile limit strength and other strength properties. Irradiation embrittlement is due to the stress-induced growth of helium bubbles on the metal grain boundary, which eventually leads to an intergranular fracture [25]. Irradiation embrittlement results in the decrease of elongation rate and the reduction of area and other ductility properties. With regard to the fatigue parameters, irradiation hardening results in the increase of the fatigue strength coefficient, while irradiation embrittlement leads to the reduction of area:(5)$$\sigma_{f}^{\prime }=\sigma_{f0}^{\prime }+\sigma_{f1}^{\prime }\left[1-exp\left(\frac{-d}{d_{\sigma_{f}^{\prime }}}\right)\right]$$
(6)$$\psi =\psi_{0}-\psi_{1}\left[1-exp\left(\frac{-d}{d_{\psi }}\right)\right]$$

The relationship between the fatigue ductility coefficient and the reduction of area is [26]:(7)$$\varepsilon_{f}^{\prime }=\ln\frac{1}{1-\psi }$$

Therefore, irradiation results in the increase of the fatigue strength coefficient and the decrease of the fatigue ductility coefficient. For reactor structures under different working conditions, fatigue life is affected by irradiation differently. From the engineering perspective, the fatigue life of the core barrel flange and hold-down spring decreases with the increase of neutron dose.

#### 2.1.3. Fatigue Life Prediction Model Considering Irradiation Effects

Utilizing Equations (3)–(7), the fatigue life prediction model considering irradiation effects can be written as follows:(8)$$\frac{\Delta \varepsilon_{t}}{2}=\frac{\sigma_{f}\left(d\right)}{2}\times {\left(2N_{f}\right)}^{b}+\varepsilon_{f}\left(d\right)\times {\left(2N_{f}\right)}^{c}\;=\frac{\sigma_{f0}+\sigma_{f1}[1-exp\left(\frac{d}{d\sigma_{f}}\right)]}{E}\times {\left(2N_{f}\right)}^{b}\;+\ln\frac{1}{1-\left\{\psi_{0}-\psi_{1}\left[1-exp\left(\frac{-d}{d_{\psi }}\right)\right]\right\}}\times {\left(2N_{f}\right)}^{c}$$

Equation (8) can also be written as:(9)$$Z=\left\{\frac{\sigma_{f0}^{\prime }+\sigma_{f1}^{\prime }\left[1-exp\left(\frac{-d}{d_{\sigma_{f}^{\prime }}}\right)\right]}{E}\right\}\times {\left(2N_{f}\right)}^{b}\;+\ln\frac{1}{1-\left\{\psi_{0}-\psi_{1}\left[1-exp\left(\frac{-d}{d_{\psi }}\right)\right]\right\}}\times {\left(2N_{f}\right)}^{c}-\frac{\Delta \varepsilon_{t}}{2}$$

The predicted value of the fatigue life can be acquired by substituting the corresponding parameter set into the limit state equation Z=0.

### 2.2. Fatigue Reliability Model Considering Cumulative Effect of Irradiation

#### 2.2.1. Cumulative Effect of Neutron Dose

During the actual operation of a reactor, the neutron dose accumulates over time; the changes over time can be calculated as follows:(10)$$d=\phi t\sigma_{s}\frac{\Lambda \bar{E_{n}}}{4E_{d}}$$
where ϕ is the neutron flux, $\sigma_{s}$ is the neutron scattering cross-section, $\bar{E_{n}}$ is the average neutron energy, $E_{d}$ is the atomic displacement threshold energy, Λ is the number of secondary atoms and t is the irradiation time.

$\phi \sigma_{s}\frac{\Lambda \bar{E_{n}}}{4E_{d}}$ is defined as the neutron injection rate $v_{d}$, so the neutron dose can be expressed as:(11)$$d=v_{d}t$$

The variable amplitude load can be divided into three types (as shown in Figure 1a–c). The first type has several constant amplitude loads with a different amplitude in a cycle period. The second type is a random load with certain regularity. The third type is a variable amplitude load that can be expressed by a functional relation.

Type 1 variable amplitude load;Suppose there are *m* constant amplitude loads with different amplitude sizes in the *j-*th cycle. The total load action time is *T_N,j_* and each constant amplitude load action time is *T_N,ij_*. The number of load cycles for the constant amplitude load is *n_ij_*. The neutron dose corresponding to the *i-*th constant amplitude load within the *j-*th load cycle is:(12)$$d\left(N_{ij}\right)=d_{0}+v_{d}\times (\overset{j-1}{\underset{j=1}{\sum }}T_{N,j}+\overset{i}{\underset{i=1}{\sum }}T_{N,ij})$$
where $d_{0}$ is the initial neutron dose.The following equations can then be simply obtained by definition:(13)$$T_{N,1}=T_{N,2}=\ldots =T_{N,n}$$
(14)$$T_{N,f}=\overset{m}{\underset{i=1}{\sum }}T_{N,ij}$$Type 2 variable amplitude load;It is usually reasonable to select a representative load history section; assuming that it is repeated, the actual load can then be decomposed into a constant amplitude load with two parameters (range and mean value) using the rainflow-counting method. The load can be dived into *m* constant amplitude loads with different amplitude sizes in the *j*-th cycle. The neutron dose corresponding to the *i*-th constant amplitude load within the *j*-th load cycle is:(15)$$d\left(N_{ij}\right)=d_{0}+v_{d}\times \overset{j}{\underset{j=1}{\sum }}T_{N,j}$$The neutron doses corresponding to each constant amplitude load within the *j*-th load cycle are equal:(16)$$d\left(N_{1j}\right)=d\left(N_{2j}\right)=\ldots =d\left(N_{mj}\right)$$Type 3 variable amplitude load.Suppose that the relation of load changes with time is known:(17)$$P_{N,ij}=f\left(T_{N,ij}\right)$$Decompose *T_N,j_* into *m* intervals and the corresponding time of each interval is:(18)$$T_{N,ij}=\frac{T_{N,j}}{m}$$The load amplitude corresponding to each interval is:(19)$$\Delta P_{N,ij}=\left|f\left(T_{N,ij}\right)-f\left(T_{N,\left(i-1\right)j}\right)\right|$$The neutron dose corresponding to the *i*-th interval within the *j*-th load cycle is:(20)$$d\left(N_{ij}\right)=d_{0}+v_{d}\times (\overset{j-1}{\underset{j=1}{\sum }}T_{N,j}+i\times \frac{T_{N,j}}{m})$$

#### 2.2.2. Irradiation Fatigue Reliability Model

After considering the cumulative effect of irradiation, the neutron dose is brought into the fatigue life model to acquire the corresponding fatigue life *N_f,ij_*:(21)$$X\left(\Delta \varepsilon_{t},d\left(N_{ij}\right),E\ldots \right)\to N_{f,ij}$$

The corresponding fatigue damage $D_{f,ij}$ is:(22)$$D_{f,ij}=\frac{1}{N_{f,ij}}$$

According to Miner’s linear cumulative damage law, the total fatigue damage is:(23)$$D_{f}=\sum \frac{n_{ij}}{N_{f,ij}}$$

Using the fatigue reliability model considering the cumulative effect of irradiation, the influence of parameter uncertainty can be quantified. The strain amplitude is selected from the working condition parameters. The fatigue strength coefficient, elasticity modulus, fatigue strength exponent, fatigue ductility coefficient and fatigue ductility exponent are selected as the uncertain parameters for the material parameters. The distribution characteristics of the strain amplitude can be fitted by using multiple sets of the simulation results. Material parameters can generally be characterized by a normal distribution with a typical value as the mean value and the coefficient of variation between 0.01 and 0.05. The failure criterion is fulfilled when the total fatigue damage reaches 1 for the first time, the pertinent function can then be defined as:(24)$$Z_{f}=1-f_{D_{f}}(\frac{\Delta \varepsilon_{t}}{2},E,\sigma_{f}^{\prime },\varepsilon_{f}^{\prime },b,c,d)$$

Some of the variables that affect the total fatigue damage can be considered as random variables, so $Z_{f}$ is also a random variable. According to the definition of reliability, it can be considered that $Z_{f}>0$ is in a reliable state, $Z_{f}<0$ is in a failure state and $Z_{f}<0$ is in a limit state.

The Monte Carlo method was used to generate data sets. According to the fatigue model, the total fatigue damage distribution can be obtained as:(25)$$D_{f}=f_{D_{f}}(\frac{\Delta \varepsilon_{t}}{2},E,\sigma_{f}^{\prime },\varepsilon_{f}^{\prime },b,c,d)$$

Combined with Miner’s linear cumulative damage law and the failure criterion, reliability can be defined as:(26)$$R=P\left(Z_{f}>0\right)=P(D_{f}<1)$$

The reliability at different times can be calculated by using the Monte Carlo simulation method and thus the reliability of the reactor structure during a long-term operation can be quantitatively predicted.

## 3. Case Study and Results

The core barrel flange is located between the reactor pressure vessel supporting the protruding platform, the hold-down spring and the core barrel (as shown in Figure 2). It carries almost all the weight of the whole of the reactor internals. The flange is affected by the fluctuation of cooling water temperature during the operation and it bears the variable amplitude temperature load. The core barrel flange is subjected to fatigue failure problems resulting from the actions of a constant mechanical load and variable temperature load. In addition, as a component of the reactor internals, the core barrel flange is affected by irradiation in the reactor operation process. It is necessary to carry out a fatigue reliability analysis of the core barrel flange that takes into account the irradiation effect.

### 3.1. Stress–Strain Simulation of Core Barrel Flange

When calculating fatigue life according to the fatigue model, the strain amplitude under a certain load is needed. Therefore, using ANSYS Workbench, a finite element simulation method was deployed to acquire the strain amplitude of the core barrel flange and thus provide an input for the fatigue life model [27]. The CAD model was established according to the structural dimensions and material properties of the core barrel flange. The solid 186 element was selected for automatic mesh generation and manual optimization used for necessary sites. Based on the mesh model, the loads and support conditions were set. Next, a fixed support was added to the upper contact surface of the core barrel flange. Then, a frictionless support was added between the pressure vessel supporting the protruding platform and the core barrel flange. Moreover, standard Earth gravity at the center of gravity, a line pressure of 289.1 kN/m at the contact line of the hold-down spring and a gravity load of 2450 kN for the other components internal to the contact surface of the core barrel were added. The simulation analysis was carried out after the temperature cycle load was input. Based on the deterministic simulation analysis, the uncertainty of the dimension parameters and material parameters was considered. Dimension parameters included hole depth and fillet radius (as shown in Figure 3). Material parameters included yield strength, density and Poisson’s ratio. Parameter sampling was conducted according to the distribution characteristics of uncertain parameters (as shown in Table 1). The distribution function of the total strain amplitude was obtained by multiple simulation fittings and basically conformed to the normal distribution, with a mean value of 0.4% and standard deviation of 0.017%.

### 3.2. Fatigue Reliability Analysis without Considering Irradiation

The material of the core barrel flange was 304LN austenitic stainless steel and its fatigue parameters are shown in Table 2 [28].

Under the condition of no irradiation, according to the fatigue life prediction model (Equation (3)), the deterministic fatigue life of the core barrel flange was calculated to be 1.62 × 10^4^ times (the load was applied 150 times a year, corresponding to the fatigue life of 108 years).

Further considering the parameter uncertainty, the strain amplitude was selected from the working condition parameters and its distribution characteristic was obtained by simulation fitting. The fatigue strength coefficient, elasticity modulus, fatigue strength exponent, fatigue ductility coefficient and fatigue ductility exponent, which can generally be characterized by normal distribution, were selected as the uncertain parameters for the material parameters. The reliability curve of the core barrel flange was obtained by using the Monte Carlo method with a sampling calculation of 5000 iterations, as shown in Figure 4a.

### 3.3. Fatigue Reliability Analysis Considering Irradiation Effect

The core barrel flange is affected by irradiation in the actual operation process and the neutron dose can reach 3 dpa in the 60 year lifecycle. The fatigue reliability analysis of the core barrel flange was carried out under the condition of different neutron doses; the reliability curve is shown in Figure 4b.

When the reliability was 99.78%, the fatigue life of a 3-dpa neutron dose was 74.5% lower than that acquired without considering the influence of irradiation. Therefore, it is necessary to consider the influence of irradiation in the fatigue reliability analysis of a core barrel flange.

In fact, the neutron dose accumulates with time. For the core barrel flange, the neutron dose is 3 dpa within 60 years and the corresponding neutron dose rate is:(27)$$v_{d}=\frac{d}{t}=0.05\mathrm{dpa}/\mathrm{year}$$

A smaller neutron dose was added per cycle. In order to simplify the calculation, the corresponding neutron dose accumulation of each cycle in each year can be replaced by that of the current year; the corresponding neutron dose of each cycle in the *j*-th year is then as follows:(28)$$d\left(N_{ij}\right)=d_{0}+\overset{n}{\underset{j=1}{\sum }}0.05\times j$$

The corresponding fatigue life is calculated according to the annual cumulative neutron dose and a reliability curve taking into account the irradiation cumulative effect is then obtained, as shown in Figure 4c.

When the reliability was 99.78%, the fatigue life with the irradiation cumulative effect was 24.3% lower than that acquired without the irradiation effect and higher than that acquired by considering only the irradiation injection. In the fatigue reliability analysis of a core barrel flange, considering the cumulative effect of irradiation can improve the accuracy of fatigue life prediction.

## 4. Discussion

### 4.1. Influence of Strain Magnitude on Fatigue Reliability

The test results show that irradiation hardening increases the fatigue strength coefficient and irradiation embrittlement decreases the fatigue ductility coefficient. They have opposite effects on the fatigue life. Irradiation increases the fatigue life at a low strain and decreases it at a high strain. The curves for the changes in fatigue life with neutron doses under different strain amplitudes, for a reliability of 99.78%, are shown in Figure 5.

The test results show that the fatigue life was increased by irradiation at a low strain and decreased by irradiation at a high strain. The tensile properties changed obviously when the neutron dose was less than 5 dpa and the tensile properties tended to be saturated when the neutron dose was more than 5 dpa [19,20]. The calculation results show that the fatigue life increased with the accumulation of neutron doses at low-strain amplitudes and decreased with the accumulation of neutron doses at high-strain amplitudes. The fatigue life changed obviously when the neutron dose was less than 5 dpa. The calculated results are in good agreement with the test results, which proves the validity of the irradiation fatigue reliability model.

### 4.2. Influence of Neutron Dose on Fatigue Reliability

According to the case analysis in Section 3, the fatigue life of the core barrel flange decreased with the accumulation of the neutron dose and the decrease rate of the fatigue life also gradually slowed down with the accumulation of the neutron dose. The curve describing the change in the fatigue life with varying neutron dose is shown in Figure 6 for a reliability of 99.78%.

Irradiation-assisted stress corrosion cracking (IASCC) is also one of the factors affecting the reliability of the reactor structure. When the neutron dose is low, irradiation-assisted fatigue (IAF) is the main failure mechanism. IAF and IASCC should both be considered when the neutron dose is around 5 dpa. When the neutron dose increases by a certain amount over 5 dpa, the fatigue life remains basically unchanged. Irradiation is then no longer a factor affecting the fatigue reliability. The reliability analysis should be carried out from the perspective of an irradiation-based failure mechanism (such as IASCC).

### 4.3. Influence of Material Performance on Fatigue Reliability

The elements of material performance that affect the fatigue reliability under irradiation include fatigue performance and irradiation performance. Among the fatigue performance parameters, the fatigue strength coefficient and fatigue ductility coefficient were positively correlated with the reliability and the elasticity modulus was negatively correlated with the reliability. Among the irradiation performance parameters, the irradiation coefficient for the performance parameter was positively correlated with the reliability and the irradiation model parameter was negatively correlated with the reliability. The material performance parameters with positive correlations with the reliability increased by 10% and those with negative correlations decreased by 10%. The reliability curves when material performance parameters change is shown in Figure 7.

Irradiation performance parameters only had a small influence on the reliability. Among fatigue performance parameters, the fatigue strength coefficient had a moderate influence on the reliability and the fatigue ductility coefficient and elasticity modulus had a great influence on the reliability. The reliability of the reactor structure can thus be greatly improved by modifying the material performance to increase the fatigue ductility coefficient or reduce the elasticity modulus, which provides a theoretical basis for the optimization of a reactor’s structural design.

## 5. Conclusions

The case study shows that the fatigue reliability model based on the cumulative effect of irradiation can be used to analyze the fatigue reliability of a reactor structure. The results showed that the fatigue life acquired by taking into account an irradiation cumulative effect was 24.3% lower than that acquired without considering an irradiation effect (the reliability was 99.78%). Considering the irradiation cumulative effect can improve the accuracy of the fatigue life prediction.The fatigue life increases with the accumulation of a neutron dose at a low-strain amplitude and decreases with a high-strain amplitude.The fatigue life changes obviously when the neutron dose is less than 5 dpa. When the neutron dose increases by a certain amount over 5 dpa, the fatigue life remains basically unchanged. Irradiation is then no longer a factor affecting the fatigue reliability. Reliability analysis should be carried out from the perspective of an irradiation-based failure mechanism (such as IASCC).Irradiation performance parameters only have a small influence on the reliability while the fatigue ductility coefficient and elasticity modulus have a great influence on the reliability. The reliability of the reactor structure can be greatly improved by increasing the fatigue ductility coefficient or reducing the elasticity modulus, which provides a theoretical basis for design optimization;In the future, tests should be carried out with different structural materials in order to determine irradiation performance parameters and fatigue performance parameters and thus improve the prediction accuracy.

## Figures and Tables

**Figure 1 materials-14-00801-f001:**
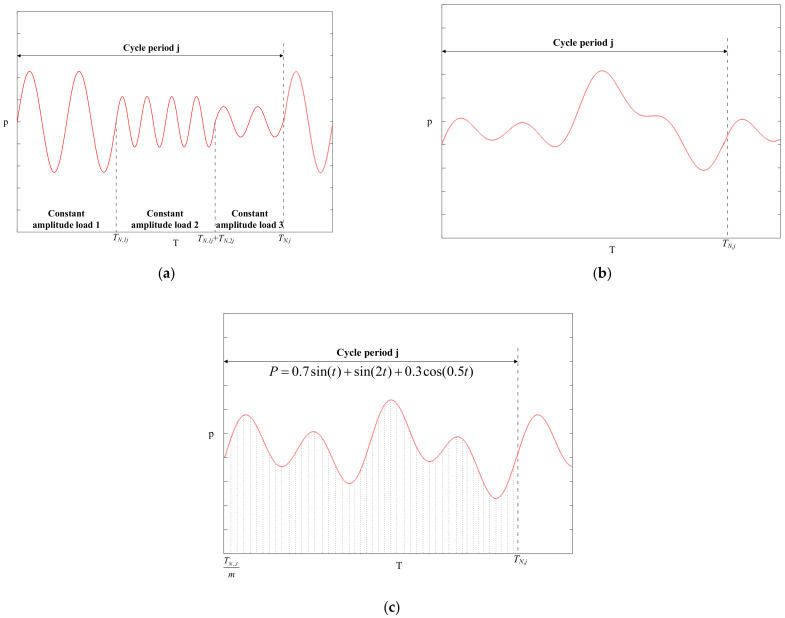
Schematic diagram of variable amplitude load: (**a**) type 1, (**b**) type 2, (**c**) type 3.

**Figure 2 materials-14-00801-f002:**
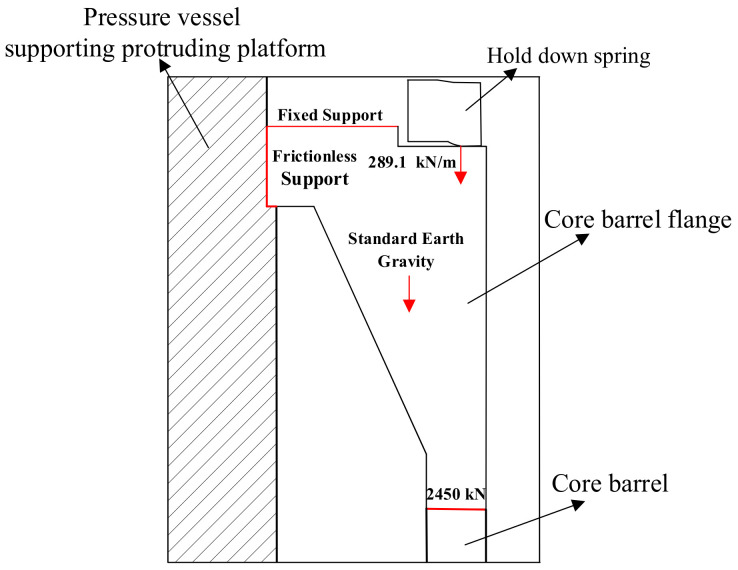
Schematic diagram of the core barrel flange.

**Figure 3 materials-14-00801-f003:**
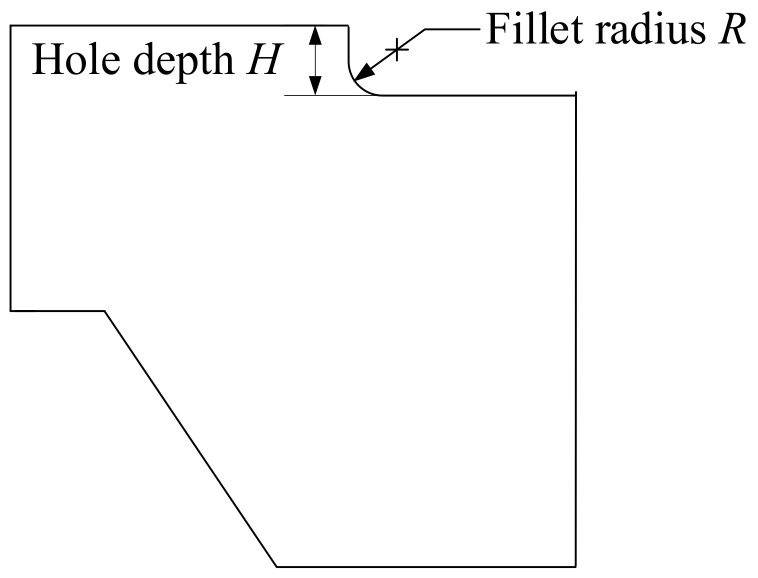
Schematic diagram of dimension parameters.

**Figure 4 materials-14-00801-f004:**
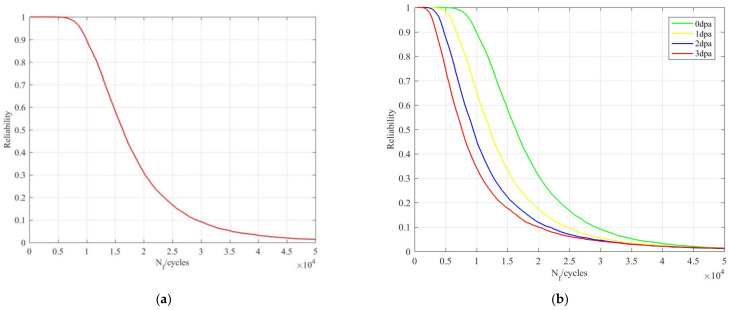
(**a**) Reliability curve without considering irradiation effect; (**b**) reliability curves of different neutron doses; (**c**) reliability curve considering cumulative effect of irradiation.

**Figure 5 materials-14-00801-f005:**
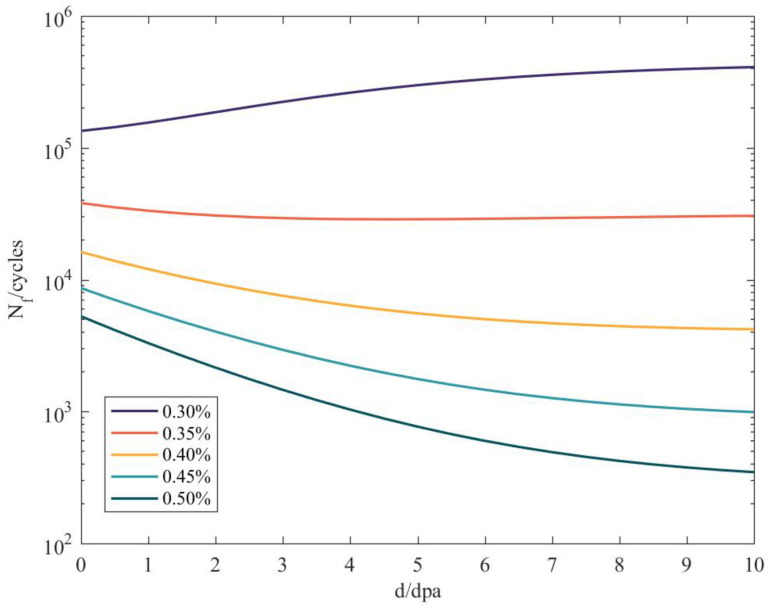
Fatigue life curves for different strain magnitudes.

**Figure 6 materials-14-00801-f006:**
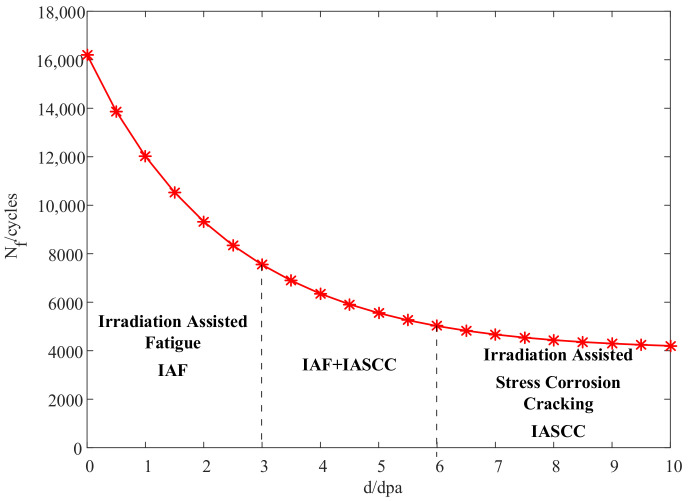
Fatigue life varying with neutron dose.

**Figure 7 materials-14-00801-f007:**
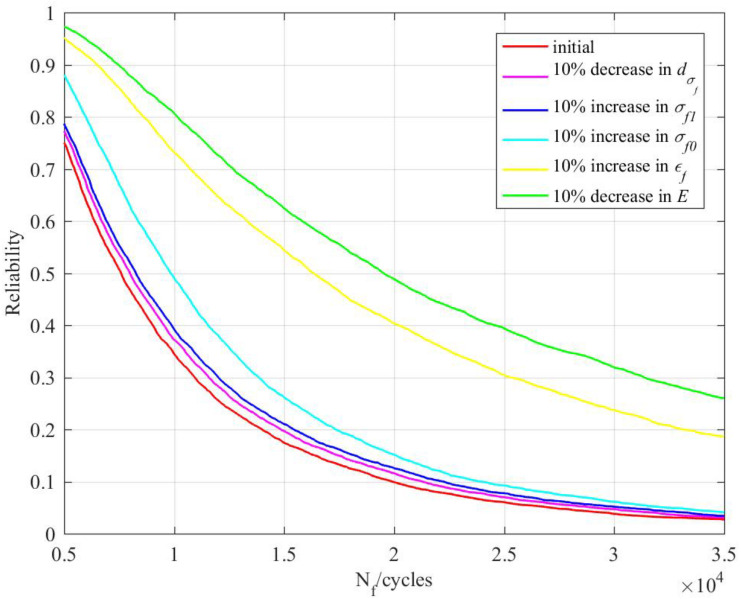
Reliability curves when the material performance parameters change.

**Table 1 materials-14-00801-t001:** Distribution characteristics of uncertain parameters.

Parameter Name	Distribution Type	Mean Value	Standard Deviation
Yield strength	Normal distribution	109 Mpa	5.45 Mpa
Density	Truncated normal distribution	7930 ${\mathrm{kg}/m}^{3}$	396.5 ${\mathrm{kg}/m}^{3}$
Poisson’s ratio	Truncated normal distribution	0.3	0.015
Fillet radius	Normal distribution	3.8 mm	0.019 mm
Hole depth	Normal distribution	14.6 mm	0.072 mm

**Table 2 materials-14-00801-t002:** Fatigue parameters of 304LN stainless steel.

Material	*b*	*c*	$\sigma_{f}^{\prime }$	$\varepsilon_{f}^{\prime }$	*E*
304LN stainless steel	−0.055	−0.529	1032 MPa	0.2422	194 GPa

## Data Availability

The data reported in this article are available on request from the corresponding author.

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
