# Peer review of "Fatigue Reliability Analysis Method of Reactor Structure Considering Cumulative Effect of Irradiation"

_materials, 2021, doi:10.3390/ma14040801_

Round 1

Reviewer 1 Report

This paper presents the influence of irradiation in the fatigue reliability model of reactor structure (core barrel flange as a case study) which gives quantification of irradiation hardening/embrittlement on fatigue performance parameters. Prediction model of the fatigue life was developed taking into account cumulative neutron dose effect on material fatigue, which can benefit reliability analysis and life prediction of specific reactor structure.  

The manuscript is informative and well structured, however, several recommendations  on how to improve the text follow (l - manuscript line number, p - manuscript page number):

 general comments (whole manuscript):

(*) The manuscript should go through additional English grammar revision, especially for missing definite/indefinite articles throughout the text on many places.

(*) Automatic numbering of the references is poor, the numbers should be given as a standard inside brackets [x], and at blank space distance. This would avoid errors such as "table 227" on line 250; should be probably "Table 2 [27]".

(*) it is customary that equations have multiplication operation with a middle dot, not x sign; you should harmonize this throughout the text

(*) insert blank between numbers and physical units, i.e. 1032MPa should be 1032 MPa

specific comments:

(l.31) "... countries that master..." should probably be "that have mastered" since you are referring to countries with developed nuclear energy   

(l.37) consider different example, more suited for your argument, such as Fukushima; the Chernobyl disaster was consequence of political-influenced low-power experiment, not something likely to occur in normal conditions. Besides that, RBMK reactors are lacking a containment building, which is (today) a standard for nuclear reactors.

(l.39) "... such as strength"  is unclear, probably should be "reactor strength"

(l.40) Reactor structure is difficult to occur..." ? something unclear

 (l.80) consider replacing "capability" of a product with "function", i.e. it is performing function, not capability

(l.159) "is" is probably surplus, i.e. neutron dose changes with time as: ...

(p.5)  consider, is possible, larger font on all figures

(l.169) consider simplifying fig. 1 caption, so not to repeat the same sentence: "Schematic diagram of variable ampl. load: (a) type 1, (b) type 2, (c) type c...

(l.171, 178, 186) consider subchapter title as: "Type x amplitude load"

(equation 14) extra blank should be removed between index numbers

(equation 18, 19) dot should be replaced with comma between index numbers

(l.181) "There" should be rephrased with "The load can be divided..."

(l.202) It is reasonable to presume "normal" distribution of material parameters for the presented model , but did you consider alternative distribution(s),  which can reflect different material behaviour (parameters) in time, relevant for real life applications? This should be addressed.

(l.228) figure 2. caption is evidently error

(l.231) comment on FEM simulation - did you use commercial code or "in-house" program? Description of the FEM code/simulation/used parameters would be useful.      

(l.237-238) First sentence is evidently wrong.

(p. 8, Table 1) "truncated" should be with capital T

(l.250) "table 227" is evidently wrong

(l.252) "...of not considering..." should be "without"

(l.307) consider rephrasing the sentence as: "The curve of fatigue....in figure 6, for reliability..."

(l.313) "mainly" should be "main"

(l.315, 349) "...increases to a certain amount" is unclear, should be "...increases to a certain amount over the 5 dpa"

(l.352) "..., and fatigue" should be "...while fatigue"

(l.362) consider rephrasing "..., so it is to further analyzes the ..."

Author Response

Dear Editors and Reviewer:

Thank you for your letter and for the reviewers’ comments concerning our manuscript entitled “Fatigue Reliability Analysis Method of Reactor Structure Considering Cumulative Effect of Irradiation.” (ID: materials-1088484). We thank the reviewers for the time and effort that they have put into reviewing the previous version of the manuscript. Their suggestions have enabled us to improve our work. We have studied comments carefully and have made correction which we hope meet with approval. We have uploaded a revised manuscript with all the changes highlighted by using the track changes mode in MS Word. Revised portion are marked in red in the manuscript. The main corrections in the manuscript and the responds to the reviewer’s comments are in attached file.

Reviewer 2 Report

Evaluation of the “Fatigue Reliability Analysis Method of Reactor Structure Considering Cumulative Effect of Irradiation”

Please check this cause do not make sense “ Due to the special working condition of irradiation,”

The conclusion are very vague and actually do not present clearly the achievement of this work

Please check for typo the entire work “ today0.”…I have noted later the typo are related to citation style please check ..

“ Reactor structure is difficult to occur over 40 stress failure because the performance parameters have large design margin” Please check this sentence do not make sense

“ has a long service life (more than 60 years)”; “obtained different (even contradictory) test conclusions “ please put a citations

“Figure 2. This is a figure. Schemes follow the same formatting.” Please check it and insert a proper caption cause is obviously that this is a Figure but what represent this Figure

You provided a long story before Figure 2 but actually will be better to show in this Figure how the loading and boundary conditions act

Which software did you have sued for simulation ???

You suggest these results “Under the condition of not considering irradiation, according to the fatigue life pre-252 diction model, the deterministic fatigue life of the core barrel flange is calculated to be 253 1.62*104 times (The load is applied 150 times per year, corresponding to a fatigue life of 254 108 years).” But not clear at all how did you achieved them !””

You shows Figure 4a-c but only indicated in text some details of Fig 4a what about other..what about interpreting these curves and results ??

Again there was presented some discussion but difficult to understand their interpretation and the link between the results presented in “results section “ and discussion section itself

For me this sentence “At present, the uniaxial stress fatigue model is used for reliability analysis, but the actual reactor structure is generally in the multiaxial stress state, so it is to further analyzes the effect of irradiation on the multiaxial fatigue behavior.” Do not make sense for this research!!! Because there is no meaning to study reliability in axial if the reactor is subjected to multiaxial stress and is not considered !!!!

Author Response

(The authors gave the same response as above.)

Round 2

Reviewer 2 Report

Thank you